# Sedentary Time and Cognitive Impairment in Patients Using Long-Term Oxygen Therapy: A Cross-Sectional Study

**DOI:** 10.3390/ijerph19031726

**Published:** 2022-02-02

**Authors:** Hiroki Annaka, Tomonori Nomura, Hiroshi Moriyama

**Affiliations:** 1Department of Occupational Therapy, Nishi-Niigata Chuo National Hospital, Niigata 950-2085, Japan; 2Graduate School, Niigata University of Health and Welfare, Niigata 950-3198, Japan; 3Department of Occupational Therapy, Faculty of Rehabilitation, Niigata University of Health and Welfare, Niigata 950-3198, Japan; nomura@nuhw.ac.jp; 4Respiratory Center, Nishi-Niigata Chuo National Hospital, Niigata 950-2085, Japan; moriyama.hiroshi.wn@mail.hosp.go.jp

**Keywords:** sedentary behavior, cognitive dysfunction, oxygen inhalation therapy, lung disease, chronic disease

## Abstract

Physical inactivity is a predictor of death in patients with chronic respiratory disease. Cognitive impairment is common among patients with chronic respiratory disease. However, the association between sedentary time and cognitive impairment in patients with chronic respiratory disease using long-term oxygen therapy is unclear. This study aimed to determine the relationship between sedentary time and cognitive impairment in patients on long-term oxygen therapy. This cross-sectional study included 96 patients with chronic respiratory disease on long-term oxygen therapy (mean age 77.3 years; female, 22%). The primary outcomes measured were sedentary time (Japanese version of the International Physical Activity Questionnaire Short Form), cognitive function (Montreal Cognitive Assessment), and dyspnea (modified Medical Research Council scale). The factors associated with sedentary time were confirmed by multiple regression analysis. The median sedentary time was 600.0 min. The median Montreal Cognitive Assessment score was 24.0 points, and 67 (70%) patients had mild cognitive impairment. In multiple regression analysis, sedentary time was associated with dyspnea (β = 0.397, *p* < 0.001) and cognitive function (β = −0.239, *p* = 0.020). This study indicates that sedentary time in patients on long-term oxygen therapy was associated with dyspnea and cognitive impairment.

## 1. Introduction

Chronic respiratory disease (CRD), including chronic obstructive pulmonary disease, interstitial pneumonia, tuberculosis, lung cancer, and nontuberculous mycobacteria, is a significant cause of death [1]. The number of deaths from CRD in 2017 was estimated to be 3.91 million worldwide [2], with the strongest predictor of death being inactivity [3]. Inactivity is caused by decreased physical activity. Previous studies have revealed several risk factors for reduced physical activity in CRD patients with mild or moderate respiratory dysfunction, such as dyspnea, high body mass index (BMI), and poor exercise capacity [4,5,6,7]. However, these factors have not reduced physical activity and inactivity in CRD patients with severe respiratory dysfunction. Understanding the factors that lead to inactivity is crucial for improving the prognosis of CRD patients with severe respiratory dysfunction.

CRD patients with severe respiratory dysfunction, which causes a decrease in the partial pressure of arterial oxygen and, eventually, respiratory failure, require long-term oxygen therapy (LTOT). LTOT improves survival outcomes and increases physical activity in CRD patients [8]. In contrast, a few previous studies have indicated that patients on LTOT have increased sedentary time, which represents inactivity, compared to non-LTOT patients with comparable respiratory dysfunction [9]. To determine the optimal treatment strategies for CRD patients with severe respiratory dysfunction on LTOT, it is essential to understand the cause of increased sedentary time in this group.

Extrapulmonary symptoms are common among patients with CRD, including heart failure, anemia, and depression [10,11]. Recently, cognitive impairment has been recognized as a comorbidity of CRD [12,13]. CRD patients have problems with specific cognitive functions, such as memory and cognitive flexibility [13,14], leading to poor adherence to treatment with inhalers and subsequent poor health and long-term hospitalization [15]. The causes of cognitive impairment in CRD patients include changes in brain structure due to hypoxemia [13,16]. A recent meta-analysis demonstrated that poor respiratory function increased the risk of cognitive impairment [17]. This finding suggests that the prevalence of cognitive impairment among patients on LTOT with severe respiratory dysfunction may be high, although this has not been directly investigated. Patients on LTOT require continuous oxygen inhalation and operate complicated equipment such as a concentrator or portable oxygen daily [18]. Operating LTOT equipment involves changing the portable oxygen cylinder, adjusting the oxygen flow rate, and handling the cannula [18]. Incorrectly operating the LTOT equipment may cause increased sedentary time due to inappropriate oxygen inhalation, impairing cognitive function; therefore, cognitive impairment may significantly lead to increased sedentary time in patients on LTOT. However, whether cognitive impairment affects the sedentary time of patients on LTOT is unknown.

Investigating the relationship between sedentary time and cognitive impairment in patients on LTOT will help elucidate the factors causing inactivity. Hence, this study aimed to investigate the relationship between sedentary time and cognitive impairment in patients on LTOT. We hypothesized that LTOT is associated with sedentary time and cognitive impairment in CRD patients.

## 2. Materials and Methods

### 2.1. Study Design and Patients

The study’s inclusion criteria were a diagnosis of CRD, as defined by the World Health Organization criteria [1], no exacerbation of respiratory function in the past two months [19], use of LTOT due to respiratory failure, and living with a housemate. Patients were excluded if they had a history of neurologic brain disease, psychiatric disorders, and relevant depressive symptoms, as determined by a Hamilton Depression Rating Scale score of ≥8 [20]. The Declaration of Helsinki was observed in the conduct of the study. All patients and their housemates provided written informed consent before study enrolment.

### 2.2. Demographics and Clinical Data

We used a cross-sectional study. The ethics committee approved the study protocol of Nishi-Niigata Chuo National Hospital (approval no. 1921) and Niigata University of Health and Welfare (approval no. 18316–191115). This study enrolled CRD patients on LTOT attending an outpatient clinic at the Department of Respiratory Medicine, Nishi-Niigata Chuo Hospital, from 1 December 2019 to 31 November 2020. All data collection was performed on the same day as an occupational therapist’s outpatient care.

### 2.3. Measurement

Sedentary time was measured by the Japanese version of the International Physical Activity Questionnaire Short Form (IPAQ), an internationally standardized questionnaire for measuring physical activity [21]. The measurement method involved collecting information from the patients about vigorous physical activity, moderate physical activity, walking, and sedentary time in the preceding week. In a pilot study of 10 patients, none of them reported vigorous physical activity, moderate physical activity, or walking; therefore, we measured only their sedentary time. Based on a previous study, we defined sedentary time as the total time spent sitting and lying in a day [21,22]. Regarding sedentary time, we asked their housemates questions about the time spent sitting at a desk, reading, or lying down to watch television in the last seven weekdays; however, this excluded sleep time. Responses were obtained in min. The IPAQ has been used in mild cognitive impairment (MCI) and CRD patients, including Interstitial pneumonia and chronic obstructive pulmonary disease [23,24,25]. The IPAQ was completed by a housemate, as previously described [23]. This questionnaire was shown to be valid in a large sample of CRD patients against accelerometers [25].

The Montreal Cognitive Assessment (MoCA) measures cognition and is a screening test to detect MCI with a maximum score of 30 points [26]. This measurement tool consists of seven domains; visuospatial and executive (points: 0–5), naming (points: 0–3), attention (points: 0–6), language (points: 0–3), abstraction (points: 0–2), delayed recall (points: 0–5), orientation (points: 0–6). MoCA has a cut-off of <26 points for MCI and has high specificity and sensitivity in patients with CRD [27].

The modified British Medical Research Council scale (mMRC) measures the severity of dyspnea in daily life [28]. It consists of five grades, from 0 to 4, with higher grades indicating limitations in daily life activities because of dyspnea. Patients with different types of CRD, including obstructive lung disease and restrictive lung disease, have other characteristics of forced expiratory volume in 1 s and vital capacity percentage depending on the individual disease. However, previous studies have confirmed the validity of the mMRC to evaluate symptom severity as an alternative method to respiratory function in various CRDs [7,28,29].

BMI was calculated using the formula: weight (kg)/height (m)^2^ and was measured using a weighing scale and a height scale.

Hand-grip strength was measured three times in the dominant hand, and the average of the three values was calculated [30]. The measurement was obtained using a digital isokinetic hand dynamometer (Model TKK5401, Grip D; Takei Scientific Instruments, Ltd., Niigata, Japan) [31].

Barthel Index is a measurement of the basic activities of daily living [32]. The score has a minimum score of 0 and a maximum score of 100 points, with a higher score indicating the ability to perform more independent basic activities of daily living. We inter-viewed the subjects’ housemates to obtain the details regarding the subjects’ life situations [33].

### 2.4. Statistical Analysis

The mean and standard deviation (SD) or the median and interquartile range (IQR) of variables such as basic information, sedentary time, MoCA: total points and points for each domain, mMRC, BMI, and hand-grip strength were presented. The normal distribution of each variable was confirmed using the Shapiro–Wilk test. Pearson’s correlation coefficient or Spearman’s rank correlation coefficient confirmed the correlation between sedentary time, age, mMRC, BMI, hand-grip strength, and MoCA. Factors associated with sedentary time were confirmed by multiple regression analysis using the forced entry method. This study determined, via a review of previous studies, that the independent variables included age, sex, BMI, type of diseases (chronic obstructive pulmonary disease, interstitial pneumonia, other) [5,6], mMRC [4,6,7], hand-grip strength [34], and MoCA [27]. Multicollinearity was confirmed using the variance inflation factors. Variables with a variance inflation factor ≥10 were excluded from the multiple regression analysis. According to multiple regression analysis, the sample size was above 70, which required ten samples for each independent variable [35]. The level of significance was set at *p* < 0.05. Statistical analysis was conducted using IBM SPSS Statistics for Windows, Version 20.0 (IBM Japan, Tokyo, Japan).

## 3. Results

### 3.1. Characteristics of Patients

Ninety-six patients who underwent LTOT participated in the study (Table 1 and Appendix A). The patients had an age of 77.3 (SD 6.6) years, and 21 patients (22%) were women. The median daily sedentary time was 600.0 min (IQR, 360.0–870.0). The median MoCA score was 24.0 (IQR, 19.3–26.0) points, and 67 (70%) patients had MCI (cut off < 26 points). The score for each domain in MoCA is shown in Table 2.

### 3.2. Correlation with Sedentary Time

The correlation between sedentary time and each variable is presented in Table 3. Sedentary time was correlated with all variables.

### 3.3. Multiple Regression Analysis

The results of multiple regression analysis with sedentary time as the dependent variable are presented in Table 4. The analysis results indicated that mMRC (β = 0.397, *p* < 0.001) and MoCA (β = −0.239, *p* = 0.020) were factors associated with sedentary time.

## 4. Discussion

This study investigated the relationship between sedentary time and cognitive impairment in patients on LTOT. Using multiple regression analysis, we found that the sedentary time of patients on LTOT was associated with mMRC and MoCA. In addition, the patients of this study were characterized by MCI, with an incidence of 70% and longer sedentary time. These findings suggest an association between sedentary time and cognitive impairment in patients on LTOT.

### 4.1. Cognitive Impairment in Patients on LTOT

This study indicated the prevalence of MCI in LTOT patients. In this study, older adults, age-matched, had a 19% prevalence of MCI [36]. The prevalence of MCI determined using the MoCA in non-LTOT patients has been reported to be 36–44% [27,37]. The prevalence of MCI in patients of this study tended to be higher.

There are several opinions regarding on cognitive function of LTOT patients [13,19,38]. Karamanli et al. [38] was compared MoCA and mini-mental state examination (MMSE) in hospitalized LTOT and non-LTOT patients. The former score is high; further, the prevalence of MCI in LTOT patients is 36.8% lower than in this study. Whereas, Mermit et al. [19] compared cognitive function in LTOT patients and non-LTOT patients after discharge demonstrated that the former had lower scores on MMSE [19]. In a recent meta-analysis, poor respiratory function was shown to increase the risk of cognitive impairment [17]. Cognitive impairment in CRD patients is affected by changes in the brain structure due to hypoxemia [13,16]. Patients with hypoxic CRD have lower cerebral perfusion and poorer neuropsychological test results than those without hypoxia [39]. Thus, patients with poor respiratory function that required LTOT in this study may progress into organic brain syndrome, eventually leading to cognitive impairment.

In addition, patients with CRD, especially those with severe respiratory dysfunction, may have white matter damage to the brain [40,41]. It is hypothesized that systemic inflammation by mediators such as tumor necrosis factor-α, monocyte chemotactic protein-1, interleukin-8, and tobacco increase oxidative stress; however, this mechanism is not fully clear [17,40,42]. Recent studies have shown that these factors cause cerebrovascular disease, which leads to cerebral ischemia [17,40,42]. This characteristic of patients with CRD, especially severe respiratory dysfunction, may contribute to cognitive impairment.

### 4.2. Characteristics of Sedentary Time in Patients on LTOT

Few studies have evaluated the sedentary time of patients on LTOT [9,43], and the sedentary time of Japanese patients on LTOT has not been clarified. The sedentary time of patients on LTOT has been reported to be 623 min [9]. This finding is consistent with the sedentary time of the patients enrolled in this study at 600 min. In contrast, older adults’ sedentary time is 330 min [44], and non-LTOT patients are 408 min [45].

The sedentary time of patients on LTOT tends to be longer than that of non-LTOT patients. A possible reason for this may be more frequent dyspnea exacerbations among patients on LTOT, which prolongs their sedentary time [4,7]. In addition, patients on LTOT are forced to wear a cannula at home and carry portable oxygen when going out. In a meta-synthesis of qualitative research studies, Cullen et al. [46] noted that LTOT equipment usage leads to incapacity because of psychological factors such as stigma and prejudice. Furthermore, Cani et al. [9] showed that the length of the cannula from the concentrator decreased mobility in patients on LTOT. Based on the findings of previous studies, the activity of patients on LTOT may be hindered by LTOT equipment. Therefore, longer sedentary time in patients on LTOT may be caused by LTOT equipment in addition to dyspnea.

### 4.3. Relationship between Sedentary Time and Cognitive Impairment in Patients on LTOT

The main finding of this study was the association between sedentary time and cognitive impairment in patients on LTOT. Currently, the causality of this association has not been clarified. Previous studies confirmed that low physical activity levels are a risk factor for cognitive impairment in patients with CRD [47]. In contrast, a large epidemiological study demonstrated that patients with MCI have low levels of physical activity [48]. Based on these observations, longer sedentary time and cognitive impairment may influence each other. However, this study had a cross-sectional design, hence, the acausal relationship cannot be established. Thus, a longitudinal study is required to verify this causal relationship.

Moreover, patients on LTOT are required to perform complicated equipment operations such as adjusting the oxygen flow rate, preparing portable oxygen, and changing the oxygen cylinder [18]. Cognitive impairment may affect a patient’s ability to operate LTOT equipment. However, this study could not clarify the effect of LTOT equipment operation on sedentary time, and we plan to investigate this in future studies.

This study was conducted during the COVID-19 pandemic. The subjects of this study and their housemates were not infected; however, they may have experienced changes in their daily routines compared to those before the pandemic. The lifestyle changes in Japan that accompany infection prevention behavior may affect the physical and mental status. Previous studies suggest that COVID-19-related activity restrictions during the pandemic can affect the physical and mental status [49]. In this study, it is unclear whether lifestyle changes affected the sedentary time and cognitive function in the subject. Therefore, the results of this study should be interpreted with caution.

Patients on LTOT may have an increased risk of cognitive impairment due to severe respiratory dysfunction. Furthermore, cognitive impairment may limit the activity of patients on LTOT. Medical staff may need to pay attention to cognitive impairment in patients on LTOT in clinical practice.

### 4.4. Limitations

This study had several limitations. First, this was a single-center study, and the sample size was relatively small. Furthermore, the patients in this study tended to be older, although this had little impact on the study results. Future multicenter studies that include patients with a wider age range should be conducted to validate our results. Second, this study did not compare the characteristics of patients on LTOT with those age-and severity-matched non-LTOT patients. Third, sedentary time was not evaluated using pedometer and accelerometer data, which are standard and objective methods to measure sedentary time in CRD patients. However, this study may lead to further investigations on the effect of exercise capacity and sedentary time. Fourth, since the patients had different types of CRD, such as obstructive lung disease and restrictive lung diseases, disease severity was measured using the mMRC. However, using objective data, such as arterial blood gas analysis, to determine the severity of respiratory dysfunction is desirable. Fifth, to avoid physically burdening patients with severe respiratory dysfunction, we did not measure their exercise capacity. In future studies, the 6-min walk test should be conducted, which will lead to a better understanding of the characteristics of the sedentary time in patients on LTOT. Sixth, it was difficult to investigate the effects of operating the LTOT equipment on sedentary time. Investigating this association may help with self-management problems and equipment operation in patients on LTOT. Seventh, the effect of the COVID-19 pandemic on the results of this study due to changes in the lifestyle of the patients is unknown. Therefore, future comparisons with studies conducted after the COVID-19 pandemic are needed. Finally, it is essential to acknowledge that the cross-sectional nature of our study limits the ability to determine a causal relationship between sedentary time and cognitive impairment in patients on LTOT. Currently, the characteristics of patients on LTOT are unclear. Nevertheless, these results can help inform future longitudinal studies to determine whether cognitive impairment is a risk factor for a longer sedentary time in patients on LTOT.

## 5. Conclusions

This study suggests that sedentary time in patients on LTOT is associated with cognitive impairment, and cognitive impairment may be a risk factor for a longer sedentary time in patients on LTOT.

## Figures and Tables

**Table 1 ijerph-19-01726-t001:** Characteristics of patients (*n* = 96).

Characteristics	Mean (Standard Deviation) or Median (Interquartile Range) or *n* (%)
Age (year)	77.3 (6.6)
Sex (male/female)	75 (78)/21 (22)
Education (year)	12.0 (9.0–12.0)
Disease	
Chronic obstructive pulmonary disease	36 (38)
Interstitial pneumonia	40 (42)
Lung cancer (after surgery)	9 (9)
Nontuberculous mycobacteria	4 (4)
Tuberculosis (inactive)	4 (4)
Other	3 (3)
Spirometry	
Forced expiratory volume in 1 sec (%)	74.7 (50.4–90.6)
Vital capacity percentage (%)	67.5 (57.9–83.7)
Long-term oxygen therapy	
Oxygen flow (L)	3.0 (2.0–4.0)
Oxygen concentrator/Liquid oxygen	91 (95)/5 (5)
Receiving pulmonary rehabilitation	11 (11)
Heart failure (Yes/No)	59 (61)/37 (39)
Skeletal muscle disorder (Yes/No)	7 (7)/89 (93)
Diabetes mellitus (Yes/No)	10 (10)/86 (90)
Hypertension (Yes/No)	29 (30)/67 (70)
Dyslipidemia (Yes/No)	10 (10)/86 (90)
Barthel Index	100.0 (90.0–100.0)
modified Medical Research Council scale	2.0 (1.0–3.0)
Body mass index (kg/m2)	21.9 (4.2)
Hand-grip strength (kg)	24.8 (9.0)
Sedentary time (min)	600.0 (360.0–870.0)
Montreal Cognitive Assessment	24.0 (19.3–26.0)
Mild cognitive impairment (Cut off < 26 points)	67 (70)

**Table 2 ijerph-19-01726-t002:** Score for each domain in the Montreal Cognitive Assessment (MoCA).

Domain (Point)	Median (Interquartile Range)
Visuospatial and executive (0–5)	4.0 (3.0–5.0)
Naming (0–3)	3.0 (3.0–3.0)
Attention (0–6)	5.0 (4.0–6.0)
Language (0–3)	1.0 (1.0–2.0)
Abstraction (0–2)	2.0 (2.0–2.0)
Delayed recall (0–5)	1.0 (0–2.0)
Orientation (0–6)	6.0 (6.0–6.0)

**Table 3 ijerph-19-01726-t003:** Correlation coefficient (*n* = 96).

	Sedentary	Age	mMRC	BMI	Hand-grip	MoCA
Sedentary	-	-	-	-	-	-
Age	0.312 *	-	-	-	-	-
mMRC	0.546 *	0.231 *	-	-	-	-
BMI	−0.233 *	−0.041	−0.233 *	-	-	-
Hand-grip	−0.423 *	−0.511 *	−0.308 *	0.295 *	-	-
MoCA	0.386 *	−0.508 *	−0.274 *	0.044	0.303 *	-

* *p* < 0.05. Abbreviations: Sedentary, sedentary time; mMRC, modified Medical Research Council scale; BMI, body mass index; hand-grip, hand-grip strength; MoCA, Montreal Cognitive Assessment. Underline: Pearson’s correlation coefficient.

**Table 4 ijerph-19-01726-t004:** Multiple regression analysis (*n* = 96).

Independent Variable	B	β	*p*-Value	95% CI		VIF
				Lower	Upper	
Age	0.782	0.019	0.851	−7.463	9.028	1.592
Sex (0: male, 1: female)	7.760	0.012	0.913	−132.376	147.895	1.753
Type of diseases (1: COPD, 2: IP, 3: Other)	−4.094	0.011	0.900	−60.645	68.833	1.246
mMRC	95.210	0.397	<0.001	52.395	138.024	1.236
Body mass index	−3.937	−0.061	0.492	−15.272	7.399	1.200
Hand-grip strength	−5.210	−0.169	0.179	−12.846	2.427	2.372
MoCA	−14.019	−0.239	0.020	−25.822	−2.217	1.575

Dependent variable: Sedentary time. Abbreviations: B, partial regression coefficient; β, standardized partial regression coefficient; 95% CI, confidence interval; VIF, variance inflation factor; COPD, chronic obstructive pulmonary disease; IP, interstitial pneumonia; mMRC, modified Medical Research Council scale; MoCA, Montreal Cognitive Assessment.

## Data Availability

The data presented in this study are available in Appendix A.

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
