# Peer review of "Sedentary Time and Cognitive Impairment in Patients Using Long-Term Oxygen Therapy: A Cross-Sectional Study"

_ijerph, 2022, doi:10.3390/ijerph19031726_

Round 1
Reviewer 1 Report
This is an interesting study showing that sedentary time in patients on long-term oxygen therapy is associated with cognitive decline.
According to the study, data has been collected from December 1 of 2019 to November 31 of 2020, so there is a risk of the impact of Covid-19 on the overall physical and mental status of the patients as there is no historical information presented here about the previous infection with Covid-19 among the patients. If there is any record, maybe that would be possible to consider them as a hypothetical factor that could be brought here at least showing that this particular study has been conducted during the pandemic and this observed cognitive impairment could be affected by Covid-19 as well along with those of the study speculation.
The average scores of the various MoCA domains could be presented in a new table (e.g., visuospatial abilities, executive functioning, etc.)
In the discussion section "4.1": "Thus, patients with poor respiratory function that re- 186 quired LTOT in this study may progress into organic brain syndrome, eventually leading 187 to cognitive impairment."; after this sentence, it would be better to provide more evidence on the possible neurobiological mechanisms for the observed cognitive dysfunction as discussed above regarding changes in the brain structure due to hypoxemia.
This issue could be mentioned in the last parts of the discussion section
Reviewer 2 Report
This is an interesting study aimed to clarify the relationship between sitting time and cognitive dysfunction in patients with chronic respiratory diseases under long-term oxygen therapy. The conclusion of the study is of very high social/global importance and should be available for readers (in scientific and non-scientific fields) to study and apply in daily lives.
Abstract:
This study indicates that sedentary time in patients on long-term oxygen therapy was associated with cognitive impairment.
-> The results of this study were also related to the mMRC. Therefore, please state that the mMRC was also relevant.
Methods:
- In Table 1, heart failure was mentioned, but were chronic diseases other than heart disease also included in the study? For example, orthopedic diseases
- You should provide evidence for all measurement instruments that were used within the data collection of your study. Right now, none is presented on hand-grip, Barthel index.
- The independent variables were age, mMRC, BMI, hand-grip strength, MoCA, and sex.
-> Why did you choose these items as independent variables?
Was respiratory disease related to sedentary time?
You need to report the rationale for the independent variables included in the methods section.
Results:
- The normal distribution item should be changed to mean and standard deviation.
Discussion:
- Few studies have evaluated the sedentary time of patients on LTOT,
-> You should show the references.
